

# Rethinking the lake trophic state index

Farnaz Nojavan A.[1], Betty J. Kreakie[2], Jeffrey W. Hollister[2] and Song S. Qian[3]

[1] ORISE, Environmental Protection Agency, Office of Research and Development, Center for Environmental Measurement and Modeling, Atlantic Coastal Environmental Sciences Division, Narragansett, RI, United States of America

[2] Environmental Protection Agency, Office of Research and Development, Center for Environmental Measurement and Modeling, Atlantic Coastal Environmental Sciences Division, Narragansett, RI, United States of America

[3] Department of Environmental Sciences, The University of Toledo, Toledo, OH, United States of America

## ABSTRACT

Lake trophic state classifications provide information about the condition of lentic ecosystems and are indicative of both ecosystem services (e.g., clean water, recreational opportunities, and aesthetics) and disservices (e.g., cyanobacteria blooms). The current classification schemes have been criticized for developing indices that are single-variable based (vs. a complex aggregate of multi-variables), discrete (vs. a continuous), and/or deterministic (vs. an inherently random). We present an updated lake trophic classification model using a Bayesian multilevel ordered categorical regression. The model consists of a proportional odds logistic regression (POLR) that models ordered, categorical, lake trophic state using Secchi disk depth, elevation, nitrogen concentration (N), and phosphorus concentration (P). The overall accuracy, when compared to existing classifications of trophic state index (TSI), for the POLR model was 0.68 and the balanced accuracy ranged between 0.72 and 0.93. This work delivers an index that is multi-variable based, continuous, and classifies lakes in probabilistic terms. While our model addresses aforementioned limitations of the current approach to lake trophic classification, the addition of uncertainty quantification is important, because the trophic state response to predictors varies among lakes. Our model successfully addresses concerns with the current approach and performs well across trophic states in a large spatial extent.

## INTRODUCTION

Lake trophic state has become an invaluable tool for lake managers and researchers, and therefore demands due diligence to ensure that the statistical methods and results are robust. Lake trophic state is a proxy for lake productivity, water quality, biological integrity, and fulfillment of designated use criteria (*Maloney, 1979*; *USEPA, 1994*). Recreation, habitat and species diversity, property and ecological values are closely related to lake water quality (*Keeler et al., 2015*; *Leggett & Bockstael, 2000*). Hence, monitoring water quality is integral to the management of the eutrophication and productivity of lakes. In fact, the Clean Water Act requires that all US lakes be classified according to trophic status in order

Corresponding author
Farnaz Nojavan A.,
nojavan-asghari.farnaz@epa.gov,
f.nojavan@gmail.com

to provide insight about overall lake quality (*USEPA, 1974*). Trophic state can be used both as a communication tool with the public and a management tool to provide the scientific accord of eutrophication and character of the lake.

Given its broad applicability and long history, it is important to periodically review and update the methods used to calculate trophic state. The concept of trophic state, originally proposed by *Naumann (1919)*, is based on lake production and quantified by an estimation of algal biomass due to their impacts on a lake's biological structure. *Naumann (1919)* emphasized a regional approach to trophic state due to inter-regional variation in lake production. Trophic state has been formulated using various indices, the most well known was created by *Carlson (1977)*. Building on his work, others have developed numerous classification schemes which vary considerably in their approach to classification, variable selection, and category counts. In short, existing approaches are either single parameter, difficult to scale, or use classification approaches that could be improved.

Single parameter trophic state indices are based on the biological condition of a lake which is the result of lake productivity affected by multiple factors such as nitrogen, phosphorus, and other chemical variables along with light, temperature, and other physical variables. Many of these use nutrient concentrations, nutrient loading, algal productivity, algal biomass, and hypolimnetic oxygen depletion rate (for an extensive review see *Carlson & Simpson (1996a)*). A single parameter index cannot differentiate trophic state from its predictors (*Carlson & Simpson, 1996a*). The goal of developing a trophic state indicator should be to link a lake's trophic status to the main causes of its productivity, which suggests the need for a multi-parameter index.

Traditional multiparameter index approaches view trophic state as a complex response caused by interaction among various physical, chemical, and biological factors. These approaches use relevant combination of causal factors usually through definition of sub-indices and integrating the sub-indices to calculate a final index (*Carlson & Simpson, 1996a*; *Brezonik, 1984*). More recently, researchers have proposed including expert panels with measures of water quality (*Parparov et al., 2006*; *Parparov et al., 2010*). These approaches promise to more closely link ecology measurements with uses of the water bodies (e.g., drinking, recreation, etc.), yet they are challenging to apply to a large number of lakes as they require convening panels to determine how multiple parameters interact.

Classification procedures also differ greatly; some indices are quantitative and continuous, whereas others are qualitative and discrete. A continuous index accommodates trophic changes along a production gradient; however, these are often discretized for reasons of convenience and ease of communication. A discrete index classifies lakes into a small number of categories resulting in loss of information on position across the trophic continuum and lack of sensitivity to changes in predictor variables. Lakes have a large degree of variability in their response to a given variable, like nutrient concentrations, and this leads to uncertainty in the trophic response. Hence, trophic state should be formulated in probabilistic terms to quantify this uncertainty.

This paper addresses the aforementioned critiques by developing a proportional odds logistic regression (POLR) model to classify lake trophic state. The proposed model builds upon the existing trophic status classification as a starting point and reassesses the trophic
state index development and classification methods; hence, "rethinks" the lake trophic state classification and index. The model contributes to literature on trophic state in several ways. First, it generates an index that is multi-variable by using Secchi depth, elevation, total nitrogen concentration, and total phosphorus concentration. Second, the developed index is continuous and thus captures a given lakes position along the trophic continuum. Third, the index classifies lakes in probabilistic terms. Finally, while it is critical to locate a lake across trophic continuum, it is not economically feasible to monitor all lakes by conventional sampling techniques. We extend the developed POLR model to allow prediction of the trophic state of all lakes, even not extensively sampled ones. The draft extended application is available on PeerJ pre-prints.

## MATERIAL AND METHODS

### Data and study area

We used data from the United States Environmental Protection Agency's 2007 National Lakes Assessment (NLA), the National Land Cover Dataset (NLCD), and lake morphometry modeled from the NHDPlus and National Elevation Data Set (*USEPA, 2009*; *Homer et al., 2004*; *Xian, Homer & Fry, 2009*; *Hollister & Milstead, 2010*; *Hollister, Milstead & Urrutia, 2011*; *Hollister, 2014*; *Hollister & Stachelek, 2017*). Ancillary data, such as the Wadeable Streams Assessment ecological regions, is also included in the NLA (*Omernik, 1987*; *USEPA, 2006*) and is used in the extended application (see *Nojavan et al, 2019*). The sampling population included all permanent non-saline lakes, reservoirs, and ponds within the 48 contiguous United States with a surface area greater than 4 hectares and a depth of greater than 1 meter, omitting the Great Lakes. A Generalized Random Tessellation Stratified (GRTS) survey design for a finite resource was used with stratification and unequal probability of selection, resulting in over 1,000 lakes sampled across the continental United States during the summer of 2007 (Fig. 1). The source code for data pre-processing and the resultant data are available on GitHub repository https://github.com/usepa/rethinking_tsi (*Nojavan et al., 2017*).

### Statistical methods

We developed a proportional odds logistic regression (POLR) to predict lake trophic state using Secchi disk depth, elevation, total nitrogen concentration (N), and total phosphorus concentration (P). The predictors in the POLR model were selected from *in situ* and universally available GIS variables using random forest models. Our modeling work flow was as follows:

1. Variable selection using Random Forest Model: Develop a random forest model, using R's randomForest package (*Liaw & Wiener, 2002*), with 5000 trees using all variables (*in situ* and universally available GIS variables) to identify the best predictor variables for lake trophic state.
2. Develop the POLR model using R function bayespolr from package arm (*Gelman et al., 2013*) and the outputs from previous step.
3. Assess the performance of the POLR model using a hold-out validation method (90% training set, 10% evaluation set).

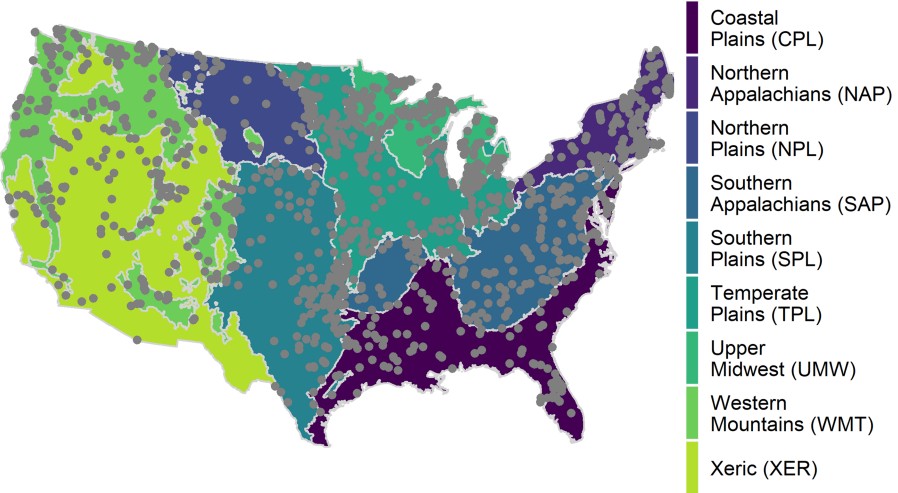

Coastal Plains (CPL)

Northern Appalachians (NAP)

Northern Plains (NPL)

Southern Appalachians (SAP)

Southern Plains (SPL)

Temperate Plains (TPL)

Upper Midwest (UMW)

Western Mountains (WMT)

Xeric (XER)

**Figure 1** **Map of the distribution of National Lakes Assessment sampling locations.** Wadeable Stream Assessment (WSA) ecoregions are also depicted in the map; the data is used in the extended application (see *Nojavan et al., 2017*). Areas in an ecoregion have similar landform and climate characteristics.

### Variable selection

The goal of variable selection is to identify an optimal reduced subset of predictor variables. Here we used the results from random forest modeling as a means of variable selection. Random forest modeling is an ensemble machine learning algorithm that builds numerous statistical decision trees in order to attain a consensus predictor model (*Breiman, 2001*). Each tree is based on recursively bootstrapped data. The out-of-bag (OOB) data, cases left out of the sample, provides an unbiased estimation of model error and measure of predictor variable importance. Random forest was a preferable method for variable selection in this case as the potential variables had a very high degree of multicolinearity which would have proved challenging for more traditional methods such as stepwise regression.

Random forest modeling was conducted with the randomForest package in R (*Liaw & Wiener, 2002*; *R Core Team, 2016*). We developed random forest models and variable importance to select predictor variables to model trophic state. The random forest model for trophic state included *in situ* water quality data and universally available GIS data, e.g., landscape data (see *Hollister, Milstead & Kreakie (2016)* for detailed methods. We selected the minimum number of variables that provided a model with a mean square error of 0.1.

### Variable transformation

Using the central limit theorem, *Ott (1995)* demonstrates that environmental concentration variables are log-normally distributed. As such, we log-transformed total nitrogen concentration, total phosphorus concentration, and Secchi disk depth data prior to our statistical analyses. Additionally, we note that the interpretation of regression model coefficients are different when log-transformed (*Qian, 2010*). Further, all predictors in the POLR model (discussed in the following section) were centered and scaled (i.e., standardized) (*Gelman & Hill, 2007*; *Gelman, 2008*).

Centering, subtracting a constant (usually mean of the variable) from every observation, simplifies the interpretation of the intercept when predictors cannot be set equal to zero. Scaling, division of each observation by the standard deviation of the variable, also improves the interpretation of coefficients in models with interacting terms, and coefficients can be interpreted on approximately a common scale. *Weisberg (2005)* also demonstrates that centered predictors would result in uncorrelated regression model coefficients.

### Proportional odds logistic regression model

The response variable, lake trophic status, is a categorical variable that can take on four values: (1) oligotrophic, (2) mesotrophic, (3) eutrophic, and (4) hypereutrophic. The four categories are separated by cutpoints (thresholds) on the continuous trophic state index. The categories are ordered across the trophic continuum with the levels of nutrient enrichment increasingly enhanced. The eutrophication process is continuous and the trophic status division is an artificial break down of a continuous index for management implications. Further, the effects of nutrient enrichment is a result of multiple factors. Any attempt to delineate categoric trophic status could be confusing when used away from lakes used to derived the index. As a result, a realistic indicator of trophic status should be a continuous and monotonic function of multiple relevant factors, which can be interpreted in terms of the traditional trophic status classification using probability of trophic status assignment to represent uncertainty. The Proportional Odds Logistic Regression (POLR) model is a statistical model that fits this need. The POLR model, a generalized linear modeling technique, has been used to account for the ordered categories of the response variable (*Gelman & Hill, 2006*). The ordered categorical response variable, lake trophic status, can be described, in its simplest form, as follows:

$$logit(Pr(\text{lake trophic status} > k)) = \text{trophic state index} - c_k \tag{1}$$

On the right side of the Eq. (1), we have the trophic state index which will be calculated using a simple linear regression (we will explain it shortly). The parameters $c_k$, cutpoints, are $k = 1, 2, 3, 4$, where $k$ is the level of an ordered category with 4 levels (oligotrophic, mesotrophic, eutrophic, and hypereutrophic), with $c_1 = 0$, and $c_2 = c_{Oligo|Meso}$, $c_3 = c_{Meso|Eu}$, and $c_4 = c_{Eu|Hyper}$.

On the left side of the Eq. (1), $Pr(\text{lake trophic status} > k)$ means the probability of a lake's trophic state being higher than k. For example for $k = 2$ (i.e., mesotrophic), $Pr(\text{lake trophic status} > 2)$ means the probability of a lake's trophic state being eutrophic or hypereutrophic. The logit means log odds, i.e., $logit(p) = log(p/(1-p))$.

The trophic state index is calculated using a simple linear regression with Secchi disk depth (SDD), elevation, nitrogen, and phosphorus as its predictors. Associated with each predictor is a coefficient $\alpha$. Figure 2 depicts all the elements of the POLR model. Mathematically, the POLR model is set up as follows, which is equivalent to Eq. (1):

$$y_i = \begin{cases} Oligotrophic & \text{if } z_i < c_{Oligo|Meso} \\ Mesotrophic & \text{if } z_i \in (c_{Oligo|Meso}, c_{Meso|Eu}) \\ Eutrophic & \text{if } z_i \in (c_{Meso|Eu}, c_{Eu|Hyper}) \\ Hypereutrophic & \text{if } z_i > c_{Eu|Hyper} \end{cases} \tag{2}$$

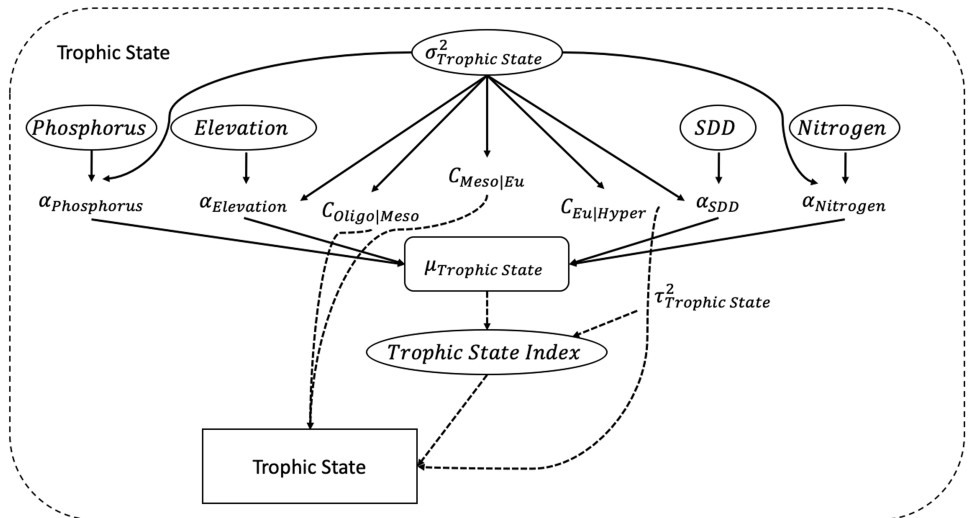

**Figure 2  Directed Acyclic Graphical (DAG) model.** The figure depicts the developed POLR model with its four predictors of secchi disk depth (SDD), elevation, nitrogen, and phosphorus.

$$z_i \sim logistic(XA, \tau^2)$$

$$
\begin{aligned}
XA = \ &\text{Secchi Disk Depth}_i \times \alpha_{SDD} \\
&+ \text{Phosphorus}_i \times \alpha_{Phosphorus} \\
&+ \text{Nitrogen}_i \times \alpha_{Nitrogen} \\
&+ \text{Elevation}_i \times \alpha_{Elevation}
\end{aligned}
$$

where:

$X = $ Design matrix of predictors

$A = (\alpha_{SDD}, \alpha_{phospurous}, \alpha_{Nitrogen}, \alpha_{Elevation})$, vector of coeeficients

$c_k = (0, c_{Oligo|Meso}, c_{Meso|Eu}, c_{Eu|Hyper})$, cutpoints or thresholds

$\tau^2 : scale parameter$

The two adjacent cutpoints and *XA* are used to classify the response variable. The cutpoints and coefficients are estimated simultaneously using maximum likelihood. For example, the probability of a lake's trophic status, being for example eutrophic, can be calculated using Eq. (1) as follows:

$Pr($lake trophic status $=$ eutrophic$) = Pr($lake trophic status $>$ mesotrophic$) - Pr($lake trophic status $>$ eutrophic$).$

The trophic status is eutrophic when the $Pr($lake trophic status $=$ eutrophic$)$ is the highest in comparison to the probability of other trophic categories, which happens when $c_{Meso|Eu} < $ trophic state index $< c_{Eu|Hyper}$ , where $c_{Meso|Eu}$ and $c_{Eu|Hyper}$ are cutpoints envisaged as unknown points on the trophic state index continuum.

### Model evaluation

The NLA 2007 used existing methods for trophic state classification based on chlorophyll *a*, nitrogen, and phosphorus. *Carlson (1977)* developed trophic state index using three

variables (i.e., chlorophyll *a*, nitrogen, and phosphorus); however, depending on which variable is used to calculate the trophic state index and categories, the resulting index and categories can be different. The reasons behind the lack of agreement between the common classification methods is discussed in detail by *Carlson & Havens (2005)*. For this reason, we avoided deviations in our evaluation data by only using 10% of the consistently classified lakes across the three trophic state classification methods (i.e., chlorophyll *a*, nitrogen, and phosphorus) as our validation set. We used a hold-out validation method where we divided the data into two subsets: a training set, used to develop the predictive model, and a validation set, used to assess the performance of the developed model. This is similar to the concept of "posterior predictive model checking" described by *Gelman et al. (2014)*, where the model predictions are being compared to the observed data looking for any discrepancies. We decided to use this approach, as opposed to validating the model with a new data set, as a comparable dataset was not available during the model development process. We evaluated the model using balanced accuracy, the average of the proportion of correct predictions within each class individually, and overall accuracy, the proportion of the total number of correct prediction.

## RESULTS

### Variable selection: random forest

The random forest models provided estimates of variable importance for trophic state and the results are reported in Fig. 3. The number of variables for each response variable was decided using the variable selection plots (Fig. 4) which show model mean squared error as a function of the number of variables. We used seventy predictor variables in the random forest model for trophic state and it indicated the best representation of trophic state classification could be achieved using four variables, adding more than four variables had incremental ($< 0.1$) impact on root mean square error. The four most important variables were turbidity, total phosphorus, total nitrogen, and elevation. The NLA uses Secchi disk depth as a measure of water clarity and, hence, we used it as a proxy for turbidity, as it is cheaper to measure and readily available for most lakes. Initially, we hypothesized that lakes at different elevations would have distinct ecological responses to similar nutrient inputs. These different responses may be related to geology, climate, or hydrology. Regardless we included elevation as proxy for an unmeasured aspect of lake dynamics.

### Proportional odds logistic regression model

The trophic state index is calculated as: $TSI = -1.69 \times$ Secchi Disk Depth$_i$ + $0.69 \times$ Nitrogen$_i$ + $0.55 \times$ Phosphorus$_i$ $-0.56 \times$ Elevation$_i$. The classification rules, based on cutpoints, are described below:

$$y_i = \begin{cases} Oligotrophic & \text{if } z_i < -3.36 \\ Mesotrophic & \text{if } z_i \in (-3.36, -0.18) \\ Eutrophic & \text{if } z_i \in (-0.18, 2.62) \\ Hypereutrophic & \text{if } z_i > 2.62 \end{cases} \tag{3}$$

$$z_i \sim logistic(TSI, 1)$$

Nojavan A. et al. (2019), *PeerJ*, DOI 10.7717/peerj.7936
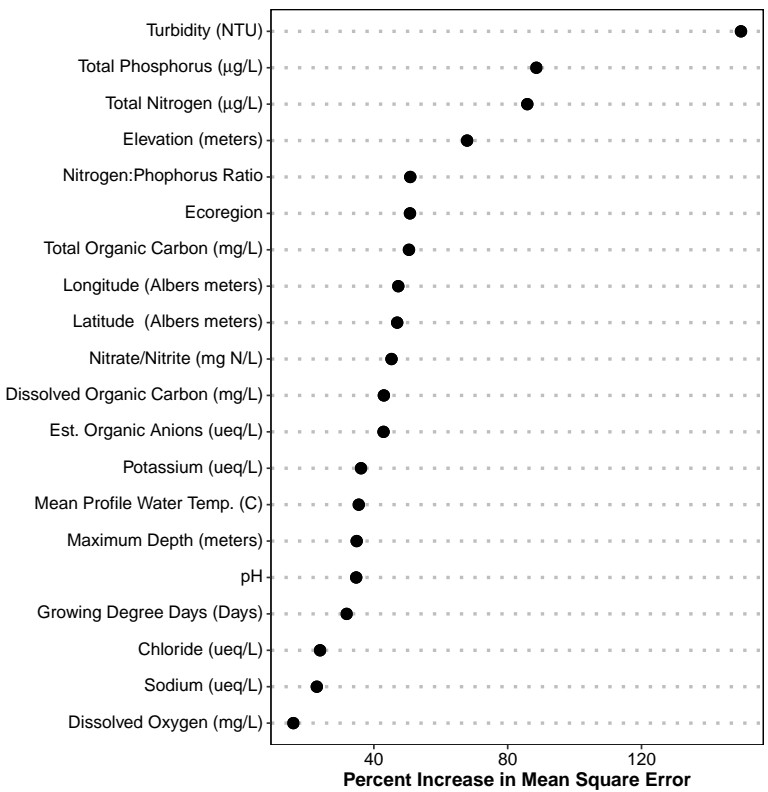

**Figure 3** **Random Forest model output for POLR model predictors.** Importance plot for all variables. Percent increase in mean squared error is shown. Higher values of percent increase in mean squared error indicates higher importance.

    The resulting POLR model has three cutpoints and four slope coefficients (Table 1). Figures. 5 and 6 summarize the model uncertainty. Figure 6 shows the probability of assigning trophic state to oligotrophic is high (the solid black line) when the trophic index is low. On the other extreme, the probability of assigning trophic state to hypereutrophic is high (the dotdash black line) when the trophic index is high. For values between the two extremes of the trophic state index, the probability for one category is highest while for the other categories the probability is not zero. The POLR model returns four probabilities associated with each trophic state as opposed to one fixed classification (Fig. 6). Therefore, for a misclassified lake the model calculates the probability for all classes. This is extremely important as lake misclassification has management implications. The POLR model output is four probabilities; each class (ologitrophic, mesotrophic, eutrophic, and hyper-eutrophic) has a probability. Hence, a manager will not just get a classification for a specific lake but also a probability for the classification. If the probabilities of two classes are close the manager knows there is a higher chance of the lake being mis-classified or just simply the lake is on the border of oligotrophic and mesotrophic. Further, unlike previous classification schemes, the proposed model keeps the continuous index as well as the discretized classes. Hence, for management applications the developed model locates

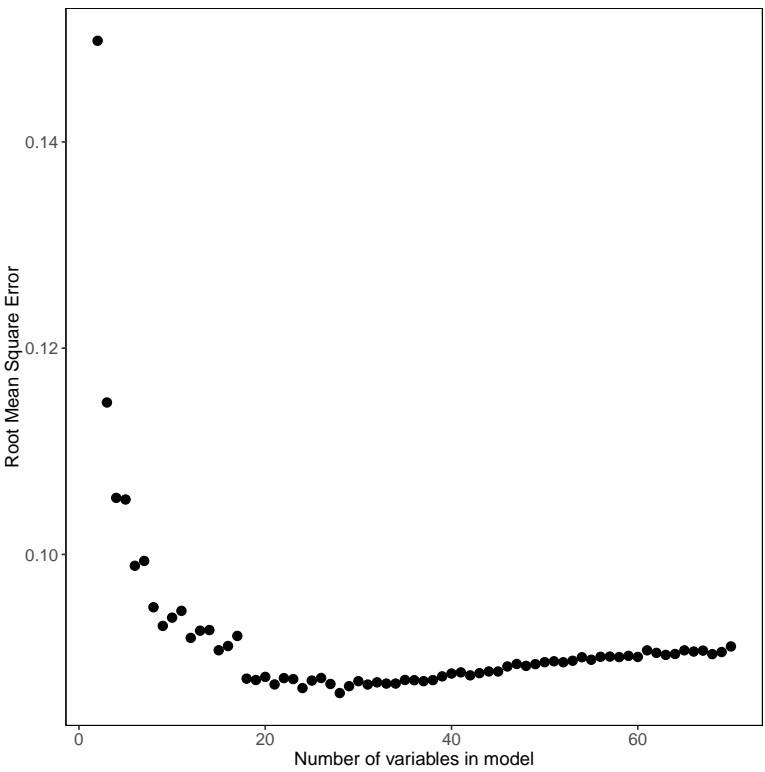

**Figure 4** **Random Forest model's output for POLR model.** The model mean squared error as a function of the number of variables is shown.

**Table 1** **Estimated POLR model coefficients and standard errors.**

|  |  | Mean | Std. Error |
|---|---|---|---|
|  | $\alpha_{Secchi\ Disk\ Depth}$ | −1.69 | 0.13 |
| *Slope* | $\alpha_{Nitrogen}$ | 0.69 | 0.13 |
| *Coefficients* | $\alpha_{Phosphorus}$ | 0.56 | 0.14 |
|  | $\alpha_{Elevation}$ | −0.56 | 0.08 |
|  | $C_{Oligo|Meso}$ | −3.36 | 0.15 |
| *Cutpoints* | $C_{Meso|Eu}$ | −0.18 | 0.09 |
|  | $C_{Eu|Hyper}$ | 2.62 | 0.13 |
| *Scale Parameter* | $\tau^2_{Trophic\ State\ Index}$ | 54.28 | 14.41 |

the lake along the trophic continuum (continuous index) and quantifies the probability for the assigned classification. Figures 5 and 6 illustrate that the proposed trophic index and classification method are now a continuum with quantified probability, hence suggesting a modified eutrophication scale that captures the inherent variability of eutrophication.

There are two ways to calculate the accuracy of a classifier: 1- The overall accuracy: the overall accuracy simply measures the number of correct classifications the classifier makes. The developed TSI model's overall accuracy is 0.68 and 2- The balanced accuracies:

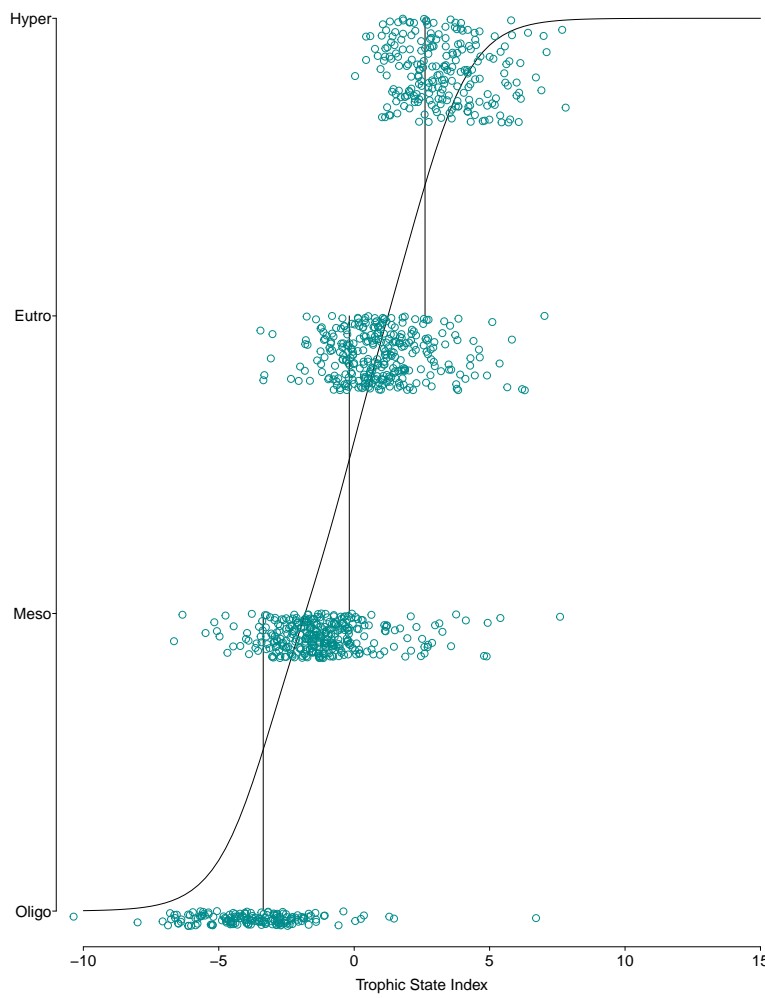

**Figure 5 Graphical presentation of the POLR model.** The *x*-axis is the trophic state index, the *y*-axis is each lake's trophic state, vertical lines show estimated cutpoints, and the curve shows expected trophic state as estimated using ordered logistic regression.

the balanced accuracies avoid inflated performance estimates on imbalanced datasets by averaging the sensitivity (true positive rate) and specificity (true negative rate). The developed TSI model's balanced accuracies are 0.93, 0.83, 0.72, 0,73 for oligotrophic, mesotrophic, eutrophic, and hypereutrophic classes, respectively. The aforementioned accuracies are calculated based on the confusion matrix. Table 2 shows the confusion matrix for the POLR model. Each element of the confusion matrix is the number of cases for which the actual state is the row and the predicted state is the column.

## DISCUSSION

### Bayesian updating and model accuracy

The Bayesian nature of the model allows us to update the model in two different ways. First, the model can be used to derive informative priors of the model parameters to combine

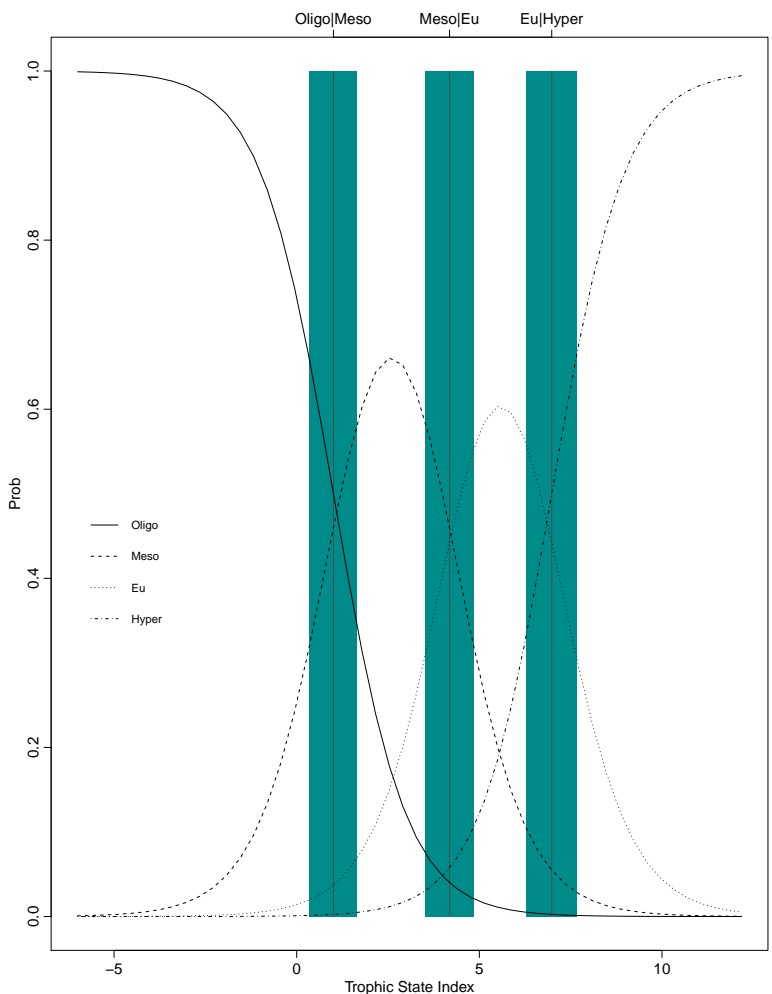

**Figure 6** **Graphical presentation of the POLR model.** The *x*-axis is the trophic state index, the *y*-axis is the probability of being classified into one of the four trophic state classes, and the vertical lines and blue bars are the cutpoints ± one standard error.

**Table 2** **Confusion matrix for POLR model.** Each element of the matrix is the number of cases for which the actual state is the row and the predicted state is the column.

|          | Oligo | Meso | Eu | Hyper |
|----------|-------|------|-----|-------|
| Oligo    | 7     | 1    | 0   | 0     |
| Meso     | 1     | 14   | 9   | 2     |
| Eu       | 0     | 0    | 16  | 8     |
| Hyper    | 0     | 1    | 4   | 10    |

with future NLA data sets (temporal model updating). More importantly, because the model is based on a cross-section of lakes, we expect that the model is not directly relevant for assessing an individual lake's trophic status. Instead, the model should be used to summarize the "average" contributions of each component variables to the national

aggregated trophic index. Such an index can be used to serve as a reference for developing individual lake index. Statistically, the aggregated national index can be used as the center of a shrinkage estimator (*Qian, Stow & Cha, 2015*). That is, the current POLR model is a prior for individual lakes. We can develop a preliminary prediction of their trophic status if a lake was not previously classified. Additional lake-specific data will enable the development of lake-specific index model.

The presented model quantifies lake trophic state and the uncertainty around it. The trophic state quantification can help in assessing lake ecological state before and after restoration. Additionally, a key symptom of eutrophication is cyanobacteria dominance in lakes (*Conley et al., 2009*; *Hollister & Kreakie, 2016*; *Przytulska, Bartosiewicz & Vincent, 2017*). The trophic state can be used as a gauge to evaluate how prone lakes are to, often toxic, cyanobacteria blooms. The uncertainty quantification helps express the resisting response of cyanobacteria to variation of phosphorus and nitrogen.

The POLR model has an overall accuracy of 0.68, yet this measure of performance fails to capture whether our stated goals were satisfactorily achieved. The accuracy measure requires that we use previous categorization of lakes based on single parameter trophic state. Somewhat circular to our goals, we are relying on discretized classifications to measure the performance of our continuous probabilistic predictions. We partially addressed this problem by using only lakes that were consistently categorized using the three common classification methods (i.e., chlorophyll *a*, nitrogen, and phosphorus) for evaluation data. A continuous scale better summarizes uncertainty, represented in the probability of being in a certain class (i.e., oligotrophic, mesotrophic, and etc.). In an attempt to circumvent this issue, we introduce balanced accuracy to measure performance of each trophic state. Balanced accuracy (as well as the confusion matrix) illustrates that misclassifications are more likely to be in adjacent trophic states. For example, the model only misclassifies one oligotrophic lake as a hypereutrophic lake; oligotrophic lakes are mostly misclassified as mesotrophic (see lower left Fig. 5). Further, in Fig. 6 as the trophic state index (*x*-axis) moves closer to the first cutpoint ($c_{Oligo|Meso} = -3.36$), the probability of mesotrophic (dotted black line) increases and the probability of oligotrpophic (solid black line) decreases; hence, the probability of misclassification between the two classes increases. To be clear, the intent of our model is not to accurately predict how lakes are classified currently, rather we show, that our model, while improving upon the statistical foundation for classification, will be comparable to existing trophic state classifications. Although we are presenting a novel method, the results are consistent with our intuitive and historical understanding of lake trophic state. One possible use case of the POLR model is presented in *Nojavan et al. (2017)*.

## Management implications

Eutrophication has constituted a serious problem for aquatic ecosystems during the past decades, largely due to excess nutrients associated with anthropogenic activities. Lake restoration projects aim to shift water quality of lakes to or closer to their undisturbed conditions. It is critical to quantitatively plan and assess the recovery of lakes in restoration projects. Our model has potential as a tool for nutrient management scenario analysis as

we can quantify how altering nutrients can move a lake across the trophic continuum. Further, updating the developed model, described in the following, evaluates the efficacy of restoration plans. Ecosystem managers and policy makers need tools that can help them learn from experience and enable them to manage the ecosystem as new knowledge becomes available. Several studies have called for adaptive management of eutrophication (*Rabalais, Turner & Scavia, 2002*; *Stow et al., 2003*).

## CONCLUSION

The modeling approach presented here uses a Bayesian ordered categorical regression model (i.e., the POLR model). The benefits of this approach are that it uses multiple variables to predict lake trophic state and creates a continuous trophic index. A multi-variable predictor model accounts for chemical, biological, and physical aspects of trophic state and quantifies lake trophic state across a continuum. This is important because lake trophic state is a variable that changes gradually across a gradient, yet it is important to predict where across the trophic continuum a lake falls, especially for lake restoration and management projects. The continuous trophic index helps us capture lake trophic sensitivity to changes in nitrogen and phosphorus. Additionally, the proposed model quantifies the uncertainty of lake trophic response to changes in nutrients, as the response varies from lake to lake. Lastly, the lake trophic index may also be presented as a classification (e.g., oligotrophic, mesotrophic, etc.) which facilitates organization and communication.

## ACKNOWLEDGEMENTS

The views expressed in this article are those of the author(s) and do not necessarily represent the views or policies of the US Environmental Protection Agency. Any mention of trade names, products, or services does not imply an endorsement by the US Government or the US Environmental Protection Agency. The EPA does not endorse any commercial products, services, or enterprises. We greatly thank Jason Grear, Bryan Milstead, Autumn Oczkowski, and Stephen Shivers for their helpful and constructive comments that contributed to improving the manuscript.

### Funding

The authors received no funding for this work.

### Competing Interests

The authors declare there are no competing interests.

### Author Contributions

- Farnaz Nojavan A. conceived and designed the experiments, performed the experiments, analyzed the data, contributed reagents/materials/analysis tools, prepared figures and/or tables, authored or reviewed drafts of the paper, approved the final draft.

- Betty J. Kreakie and Jeffrey W. Hollister conceived and designed the experiments, contributed reagents/materials/analysis tools, authored or reviewed drafts of the paper, approved the final draft.
- Song S. Qian conceived and designed the experiments, analyzed the data, contributed reagents/materials/analysis tools, authored or reviewed drafts of the paper, approved the final draft.

### Data Availability

The code is available at GitHub: https://github.com/USEPA/rethinking_tsi.

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
