# Peer review of "Rethinking the lake trophic state index"

_PeerJ, doi:10.7717/peerj.7936_

## Round 0.1 · original submission · Major Revisions

I have now received comments from two referees. Both have identified significant issues with the manuscript and recommend major revisions.

I invite you to carefully evaluate their recommendations and respond to them on a point-by-point basis. In cases where you wish to refute a point, I’d like you to consider that both reviewers are quite well known and well-published in the area of trophic indices. If they have had difficulty understanding as experts, then our general readership will likely have even greater difficulty. Thus, rather than simply refuting the point to the Reviewers in a reply, I’d encourage you to incorporate clarifications and explanations into the manuscript itself.

Among the more critical issue that need to be addressed:

1) A major element of the work is the statistical methods employed. However, as noted by Reviewer 1, these are not well introduced, explained, nor rationalized. A reader who is unfamiliar with Bayesian ordered categorical regression or proportional odds logistic regressions is, thus, unlikely to understand the advantages of these methods, and will find it hard to follow the Results and Discussion. Clearer introduction and methodological explanations are needed.

2) Reviewer 1 expresses some concerns about the literature background, which I think are justified. By my reading, some of the Parparov references have addressed the same limitations as the present manuscript. Has their work helped things? What are the shortcomings? A good literature review should consider this body of work.

3) One of Reviewer 1’s key concerns is that the new index focuses on differences among lakes in spatial terms, but is likely much less sensitive to changes in time. Since monitoring for change over time is a main application of TSI, this would certainly be perceived as a weakness. The authors should carefully evaluate the criticism. I also take the reviewer’s point that in terms of guiding lake remediation, of key variables identified, several are ‘fixed’ (elevation, latitude and ecoregion), so that the new index is relatively insensitive to management changes that could be made; this is surely a disappointing message and deserves discussion.

4) Reviewer 2 takes issue with definitions, specifically whether the authors have been successful in observing their criterion that: "the definition of trophic state should be differentiated from its predictors". Though somewhat philosophical, I tend to agree with the Reviewer’s perspective. And of course, if the “trophic state is based on the biological condition of a lake” (l. 56-57), this does beg the question of what, exactly is meant by ‘biological condition’ and what (presumably biological?) measurements are needed to define it. An early and clear set of definitions in the introduction will help.

5) Reviewer 2 has some specific concerns regarding appropriateness of use of random forest approaches, jagged functions for latitude and incorporation in models of variables with non-significant regression coefficients. In each case, replies to these points could involve Discussion points in the manuscript.

I look forward to considering your revision.

Reviewer 1 ·

Basic reporting

The MS is written in a highly technical (statistical) language that the main potential audience for the this paper may find hard to understand. I would suggest expanding on the explanation of the approaches and methods you used so the information is clear to the non-statistics expert. In addition, the text seemed at times to be very brief creating unclear sentences and links between sentences (examples below). There are a number of places where clarification is required:
ln 88- why were the great lakes omitted? They would be a good test case.
ln 148- define cMeso/Eu
lns 155-156- not clear
ln156- discrepancy between what?
ln 158- The connection between "We used a hold-out..." to the previous sentences is not clear.
ln 161: which three?
ln 178: Not clear how figs. 7, 8 summarize uncertainty- please expand on this.
ln 185-189: a mixing of topics and issues and therefore not clear.
ln 195-197: again, text not sufficiently clear
ln 223-234: Please provide an explanation as to how this could be done
ln 280: please expand on this statement so it is clear to the reader "This phenomenon is also graphically explained in Fig. 7,8"

Given the extended use of trophic indexes around the world over the last several decades I found the literature references not to be entirely up to date and somewhat limited. I would for example recommend looking at the papers by A. Parparov and colleagues who have addressed the limitations of the TLI approach and proposed several different approaches. But others are also available.
Figures: The figures are not numbered in the order they are referred to in the text. In addition there are cases in which figure numbers are not given. For example, ln 173 "(Figure ??).
The figures in the supplement material are not mentioned in the supplementary text.

Experimental design

The research question is well defined and justified and can be extremely useful. As mentioned above I do have concerns as to the description of the methods as they not necessary clear to the study's main target audience since the MS is written in a highly technical manner.
My main reservation about the approach and design used in this study is that they develop an approach for determining a new trophic index that may vary spatially between lakes but not temporally over time for a given lake. And while they state that it can be used in that way and may assist lake reclamation the fact that 1 of the 4 variables is constant (elevation) reduces the sensitivity to change in the index even if change occurs in a lake.
In the extended application the situation is even more extreme as 2 out of the 3 variables are constant (latitude and ecoregion) thus changes in conditions in a lake over time will have very little impact on the overall index value.

Validity of the findings

The largest shortcoming and limitation of the approach is the use of spatially varying variables and not temporally varying variables. The consequence is that the approach can indeed be used to compare between lakes but can't be used to look at changes over time. Hence, it is not an effective tool for lake restoration, in contrast to the statement made by the authors. This is also in contrast to the statement on ln 253 as to the ability to "quantity altering nutrients can move a lake across the trophic continuum".
The development of the extended application is a very good idea as it can be used in tandem with remote sensing and thus can be used at a global scale. However, here too, the variables selected prevent using the application to see change over time as 2 of the 3 variables are constant for a given lake. Limiting the selection of the variables that can vary over time would greatly enhance the applicability of the application.
Another point of inaccuracy relates to the statement on lns 244-245: "...quantifies the uncertainty of lake trophic responses to changes in nutrients, as the response varies from lake to lake" however the response can vary within a lake and not only between lakes. Thus there are two types of uncertainty-associated with responses to nutrients- between lakes and within lakes. The within lakes uncertainty is not addressed in the MS.

Reviewer 2 ·

Basic reporting

There are a few unformatted citations and figure references, but otherwise OK.

Experimental design

It was unclear to me what was the primary goal of the analysis being described. As the authors point out on the top of page 2, "the definition of trophic state should be differentiated from its predictors." Yet, the analysis described seems to confound these issues. Are the authors trying to define a new (continuous) *indicator* of trophic state, or better predict the already-identified trophic state of a lake from other variables? The authors assert that the goal is the former, but then I don't agree that variables such as elevation, latitude, ecoregion, or %evergreen should be part of the indicator *definition*. Rather they seem to me to be *predictors* of the indicator. This would then imply that the goal of the paper is to create a better model to predict trophic state, which is quite different and less novel. I suggest that the authors be clear about their goal and better justify why they take the approach they do in light of that goal. There is some discussion along these lines in the last paragraph, but it comes too late and does not adequately clarify things.

I also suggest that the authors justify their use of random forests for variable selection. Random forests allow for highly nonlinear and non-monotonic relationships, but the final fitted model is linear. Therefore, there may not be a correspondence between the most important variables in the two stages. In fact this can be clearly seen by comparing the apparent importance of % Evergreen Forest in Figures 5 and 6 against the results in Table S1. I suggest a stepwise linear regression approach to variable selection instead.

The authors need to be clearer about the form of their predictor variables. It only became apparent to me when I saw figures S4 and S5, that they were using some jagged function of latitude, rather than a linear or smooth relation. As noted below, I don't think such a jagged function can be interpretable as a meaningful component of a trophic state index.

Validity of the findings

I agree with the authors that regression coefficients that are not statistically significant should not necessarily be ignored. However, I am concerned that some conclusions are being drawn based on insignificant relations. For example, conclusions are being drawn about the important role of watershed conditions, such as %Evergreen, but Table S1 shows that the estimated coefficients on %Evergreen are 0.00 and -0.00 for nitrogen and phosphorus, respectively, with standard errors of 0.01. This implies that %Evergreen does not have an interpretable influence.

I am also concerned about overfitting when I see Figures S4 and S5, indicating a highly jagged influence of latitude. I don't think that such a jagged dependency plot should be used to draw conclusions about the role of latitude and I suggest that a much wider smoothing window should be used.

Additional comments

I suggest that sigma and tau values be reported in Table 1. I think Figure 2 would benefit from the mu and sigma nodes having "index" or "indicator" added to their subscripts, to distinguish from the square box of trophic state at the bottom.

---

## Round 0.2 · Minor Revisions

The Revised Manuscript certainly addressed and resolved a number of issues. As you will note, the Reviewers came up with rather different recommendations.

Reviewer 1 has some excellent suggestions which should be straightforward to accommodate in a revision.

But Reviewer 2 has a good point. In moving the "Extended Application" to Supplemental Material, the issues raised by the Reviewer and Editor haven't been addressed in any substantive way. Supplemental Material is intended to be used in a number of ways, for example, to allow presentation of information useful, but not directly relevant to the manuscript. But material included there is subject to review and needs to meet the same standard as the manuscript itself.

I am prepared to reconsider the manuscript if you do one of two things:

1) Remove the Supplemental Material. This moves things forward rapidly and avoids "confusing" the presentation of the main TSI model. I think it's fine if, in your Discussion, you refer to your additional work and its general conclusions as "unpublished". Doing so should help illustrate a possible use for the model.

2) Maintain the Supplemental Material, but address the Reviewer's concerns point by point in a revision. I would undertake to get a very rapid re-review from the Reviewer.

.

Reviewer 1 ·

Basic reporting

no further comments

Experimental design

no further comments

Validity of the findings

no further comments

Additional comments

I am happy with the corrections and responses provided by the authors to my previous comments.
I do however have two minor comments:
1. lines 213-214: "This is extremely important as lake misclassification has management implications.": This is indeed correct but how does the manager know the the lake has been misclassified? In otherwords, how does the POLR approach assist the manager in this issue?
2. lines 224-225: "The developed TSI model’s balanced accuracies are 0.93, 0.83, 0.72, 0,73 for oligotrophic, mesotrophic, eutrophic, and hypereutrophic classes, respectively." How were these values calculated? I calculated different values of .875. 0.54, 0.67, 0.67. These were calculated by (number of correct cases in a category of actual state /total number of cases in category of the actual state).

Reviewer 2 ·

Basic reporting

The article is clearly written and appropriately formatted and cited.

Experimental design

1. My previous criticism continues to hold, as it has not been addressed by the authors. Moving the extended application to the SI does not fully address my concern. To paraphrase it here: It was unclear to me what was the primary goal of the analysis. The authors state that "the definition of trophic state should be differentiated from its predictors." Yet, the analysis seems to confound these issues. I am still not clear whether the authors trying to *define* a new (continuous) *indicator* of trophic state, or better *predict* the already-identified trophic state of a lake from other variables? The authors assert that the goal is the former, but then I don't see why a variable such as "elevation" should be part of the indicator *definition*. Is it reasonable to assert that a lake at a higher elevation with the same nutrient and secchi disk values should be *defined* as being of a lower trophic state?
It might be *predicted* to be, but I don't see why it should be *defined* to be. I continue to suggest that the authors be clear about their goal and better justify why they take the approach they do in light of that goal.
2. I am not fully satisfied with the response to my comment regarding random forests. There are ways to address multicollinearity with linear regression models, and having used the method in previous studies is not a meaningful justification. However, if the authors continue to feel strongly about this point, I won't insist.
3. My concerns regarding the form of the predictors have not been addressed, although I acknowledge that I did not state them clearly. I should have said: As evidenced by the partial dependency plot, the predicted value of the response variable is changing in a very jagged way in response to changes in the predictor variable. I don't think that such a jagged dependency plot should be used to draw conclusions about the role of latitude and I suggest that a much wider smoothing window should be used. This is relevant whether this information is in the main body or the SI.

Validity of the findings

All of my previous concerns regarding the validity of findings continue to apply, even though the relevant material has been moved to the SI. They are still part of the manuscript and should adhere to the same standards.

Additional comments

The authors addressed my additional comments regarding notation.

---

## Round 0.3 · accepted · Accept

In fact, you've actually come up with a third alternative to the two I'd offered. While this isn't quite what I'd envisioned, I believe this satisfies the reviewers' concerns and I think we can trust our readers critical skills.
I recommend acceptance, but will need to check with the editorial staff about the mechanics of the preprint citation to be sure we do this correctly.